# Sustainable Adaptive Cycle Pavements Using Composite Foam Concrete at High Altitudes in Central Europe

Martin Decky [1], Katarina Hodasova [1,*], Zuzana Papanova [2] and Eva Remisova [1]

1  Department of Highway and Environmental Engineering, University of Zilina, Univerzitna 8215/1, 010 26 Zilina, Slovakia; martin.decky@uniza.sk (M.D.); eva.remisova@uniza.sk (E.R.)
2  Department of Structural Mechanics and Applied Mathematics, University of Zilina, Univerzitna 8215/1, 010 26 Zilina, Slovakia; zuzana.papanova@uniza.sk
*  Correspondence: katarina.hodasova@uniza.sk; Tel.: +421-9480-66540

**Abstract:** Climate pavement adaptability is an integral part of a holistic concept of road design, construction, and pavement management. One of the possibilities for fulfilling the mentioned author's premise in sustainable cycle pavements in the cold region of Central Europe is using composite foam concrete (CFC). To establish the credibility of the design of these pavements, we objectified the correlation dependencies of average annual air temperatures and frost indexes, for altitude regions from 314 to 858 m in the period 1971 to 2020, at its height above sea level. As part of the research on the increase in tensile strength during bending of CFC, extensive laboratory measurements were carried out and validated by isomorphic models of real roads, which enabled an increase in tensile strength during bending from 0.376 to 1.370 N·mm$^{-2}$ for basalt reinforcing mesh. The research results, verified through FEM (Finite Element Method) models of cycle pavements, demonstrated a possible reduction of total pavement thickness from 56 to 38 cm for rigid pavements and 48 to 38 cm for flexible pavements.

**Keywords:** foam concrete; reinforcing basalt mesh; annual air temperature; modulus of elasticity; cycle pavement

## 1. Introduction

Higher Education Institutions have the mandate of promoting sustainability by addressing the United Nation's 2030 Agenda [1]. The Sustainable Development Goals (SDGs) are a universal set of seventeen goals and targets, with accompanying indicators, which were agreed upon by UN member states to frame their policy agendas for the fifteen-year period from 2015 to 2030 [2]. All of the objectives of Agenda 2030, emphasizing science, technology, and innovation (STI) are most welcome, but achieving desired outcomes by 2030 will require a deep understanding of how to maximize the contributions of STI [3]. Global developments in construction give sustainability a crucial role in the overall healthy functioning of society as well as the whole environment. Modern Methods of Construction (MMC) represent a response to the sustainability trend since they bring faster construction and better environmental, energy, and economic parameters [4]. Kibert [5] laid down the foundation for Sustainable Construction (SC) practice, and established SC around resource minimization and reuse, use of renewable and recyclable resources, and minimizing carbon footprint. Vanegas and Pearce [6] presented SC based on resource depletion and degradation, impact on the built environment and human health, Braungart and McDonough [7] offered the cradle-to-cradle definition, and Pulaski [8] presented a comprehensive approach towards sustainability in construction operation, Mallick and El-Korchi [9].

In the context of road adaptive design [10,11], including pavements for walking and cycling journeys, forming 20–40% of the total journeys in the European Union [12,13], it is possible to take these data into account at an early stage [14]. Other examples of the diversity of innovative solutions point to their possible use and improvement on the impact

on the living environment and sustainability already present in the pavement design and management phase [15,16]. The authors have long devoted themselves to a holistic concept of design, construction, and pavement management, of which climate adaptability is also an integral part [17,18] along with saving non-renewable resources [19,20] One of the first approaches applied was decreasing the energy consumption during production by using new energy-efficient technologies, using reclaimed materials [21,22], incorporating new types of materials [23], waste recovery [24] as well as by modifying the technology e.g., by decreasing the processing temperature (using additives or foaming technologies [25,26]). Their use has successfully shown a positive impact on non-renewable cumulative energy demand and global warming potential. However, an important requirement for the use remains to ensure the quality, durability, and service longevity of the produced transport infrastructure.

For the last 10 years, authors in the field of STI pavement materials have been mainly researching and improving the properties of foam concrete [11,21,22,27] for the purposes of its application in pavement construction. Lightweight foamed concrete (LWFC) has found its first application in building constructions as roof slopes and floor leveling, later in lightweight blocks, thermal and acoustic insulation, void filling and trench reinstatement, slope stabilization, road subbase, encapsulate bridge piers, backfilling the voids behind the tunnel lining, sports fields, soft ground engineered arresting system for airports [28,29]. LWFC from the point of pavement application has advantageous properties including low self-weight [30], which is important in refurbishment operations; it also works to lessen the loads, establish thermal insulating characteristics [31,32], promote fire resistance [33], and reduces the cost of production and thus cost during the construction and workability [33,34]. Foam concrete has a low weight (250 to 900 $kg \cdot m^{-3}$), minimal aggregate consumption (significant saving in natural resources), controlled low strength (0.3 to 5 MPa), and excellent thermal and acoustic insulation properties (thermal conductivity of 0.058 to 0.26 $W/m \cdot K$). Its specific characteristic is that it contains closed air pores that reduce its volumetric weight. FC allows also for the incorporation of recycled and secondary materials (demolition fines or conditioned fly ash).

Recently, LWFC has also begun to find applications in civil engineering structures in Central Europe, especially at higher altitudes requiring increased protection against the adverse effects of frost [35–38]. In the present article, the authors present the latest results of their research in the field by increasing the tensile strength in bending LWFC through the application of various reinforcement nets. Optimized design of the composite foam concrete pavement layer (use of basalt reinforcement mesh) in conjunction with credible data on climatic characteristics and validated FEM models enable their application in accordance with the requirements of the Slovak Road Act [39]. Until the middle of the 20th century, methods based on limited deflection [40,41] and limited shear failure prevailed in the assessment of roads [42]. However, these assumptions are not accurate, in pavement materials, stress leads to elastic, viscous, plastic, and viscoelastic deformations; therefore, the materials behave nonlinearly [43]. The finite element method (FEM) for the analysis of flexible pavements was first applied by Duncan [44], which allowed the assessment of pavements considering the nonlinear properties of pavement materials and therefore, the use of FEM is becoming popular in determining their stresses and deflections [45–47]. This article presents, for the first time in Central Europe, a FEM model of a cycle pavement using a base layer of composite foam concrete, the relevant mechanical characteristics of which were found during the 10 years of research carried out at the authors' workplace [48–50].

The paper presents the results of the research enabling sustainable, adaptive cycle pavements design (SACD) using composite foam concrete at high altitudes in Central Europe (CE). The territorial definition of the term CE may seem straightforward at first sight, but in reality, it is not. The authors, therefore, present in a separate chapter the development and the currently established perception of CE [51–54] as well as their definition of the territory of CE. The paper also provides an overview of the highest and lowest permanent settlements of CE individual countries, whereas, for the purposes of the SACD, the authors divided the altitudes of inhabited settlements in Central Europe into four categories, namely

0–300, 301–670, 671–1000, and above 1000 m above sea level. The paper considers only altitudes above 330 m, with emphasis on settlements above 670 m, while a case study for the highest permanently inhabited village in Slovakia, Demanovska Dolina with an altitude of 1109 m above sea level, is given.

## 2. Methods

In this article, priority attention is focused on the following research activities carried out by the authors:

- Statistical analyses of the evolution of relevant climatic characteristics of Central Europe between 1971 and 2020 for altitudes above 330 m a.s.l.;
- Determination of flexural strength of isomorphic models with foam concrete according to STN EN 12390-5:2020 Testing hardened concrete. Part 5: Flexural strength of test specimens [55];
- Verification of the accuracy of laboratory tests on homomorphic models of real pavement structure [41,42];
- FEM models are used to validate the modeling methodology and to create a FEM model of the pavements of interest;
- Analytical models in Central European conditions binding for design of asphalt and cement concrete (CC) pavements with reinforces FC 500 base course.

### 2.1. Territorial Definition of the Central European Area and the Range of Heights

The development of the territorial delimitation of Central Europe from 1917 to 2022 is shown in Figure 1. It presents the current delimitation of the SE [54] with redrawn boundaries from [51]. Many geographical names either lack or lose their precise meaning, and the term Central Europe is a typical example. The development of the perception of Central Europe according to different authors is also shown in Figure 1.

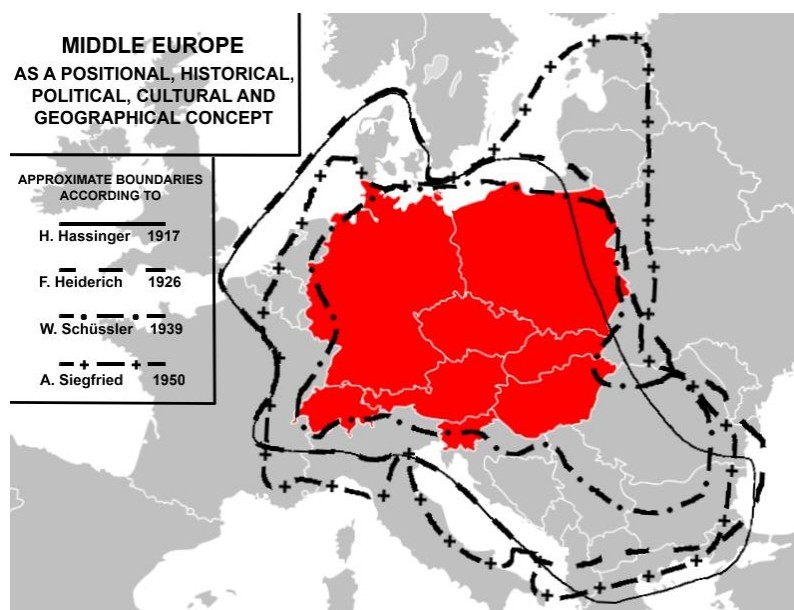

**Figure 1.** Development of the territorial delimitation of Central Europe from 1917 to 2022, redrawn boundaries from [51] to the current delimitation of the SE [54].

The location of the geographical center of Europe depends on the delimitation of Europe's borders. Particularly on whether remote islands are included to define the extreme points of Europe and on the method of calculating the result. Therefore, several places claim to host this hypothetical center: the town of Kremnicke Bane or the neighboring town Krahule, near Kremnica, in central Slovakia; the town of Rakhiv, or the town of Dilove, near Rakhiv, in western Ukraine; the town of Girija, near Vilnius, in Lithuania; a point on

the island of Saaremaa, in Estonia; a point near Polotsk, or in Vitebsk, or near Babruysk, or Lake Sho, in Belarus; a point near the town of Tallya, in northeastern Hungary (Figure 2a). The towns of Krahule and Kremnicke Bane are generally understood by Slovak authors as the geometric center of continental Europe, as well as by foreign authors [52,53].

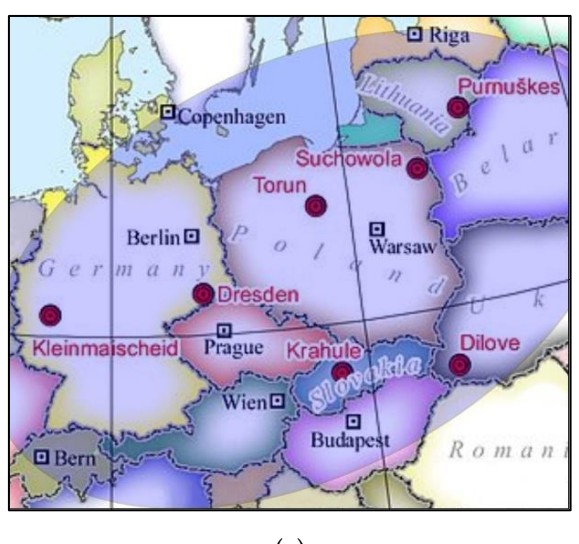

(**a**)

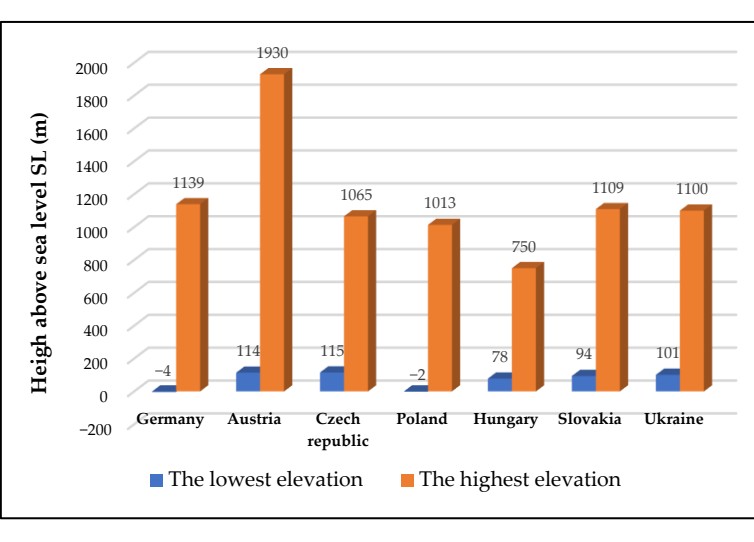

(**b**)

**Figure 2.** (**a**) The red dots on the map [56] indicate some of the cities contending for the title Center of Europe: Dilove (Rakhiv, Ukraine), Krahule (or Kremnicke Bane, Slovakia), Dresden and Kleinmaischeid (Germany), Torun and Suchowola (Poland), Bernotai, or Purnuskes (Lithuania), (**b**) the highest and lowest permanently inhabited municipalities, or places with the lowest altitude of the countries under consideration.

In this article, the highest and lowest permanently inhabited municipalities, or places with the lowest altitude of the countries under consideration are presented. The altitude is related to the definition point of the municipality, which means the center of gravity of the settlement of the town for each country was determined (Figure 2b): Germany: Neuen-dorf-Sachsenbande (4 m b.s.l.-meters below sea level)-Oberjoch (1139 m a.s.l.-meters above sea level), Austria: National Park Neusiedler See-Seewink (114 m a.s.l.)-Ober-gurgl-Hochgurgl (1930 m a.s.l.), the Czech Republic: Hrensko na Decinsku (115 m a.s.l.)-Kvilda (1065 m a.s.l.), Poland: Marzecino (2 m b.s.l.)-Ząb (1013 m a.s.l.), Hungary: Tisza (78 m a.s.l.)-Matraszentimre (750 m a.s.l.), Slovakia: Klin nad Bodrogom (94 m a.s.l.)-Demanovská Dolina (1109 m a.s.l.), Ukraine: Ruski Heivtsi (101 m a.s.l.)-Vypchyna (100 m a.s.l.).

### 2.2. Basic Criteria for the Structural Pavement Design Assessment

When designing the roadways of land roads in the Slovak Republic, there are decisive codified provisions in the road Act No. 135/1961 Coll. [39]. It is stated that the design of land roads is carried out according to the valid Slovak technical standards, technical regulations, and objectively determined results of research and development for road infrastructure. The correctness of the design of the pavement structure with its degree of reliability for the entire design period is assessed using criteria. The design of the pavement structure must meet the basic requirements criteria, which are:

- A. Road protection against the adverse effects of subsoil freezing;
- B. Ratio of flexure tensile strength and critical bending stress in asphalt or cement-bound pavements layers.

The design of the pavement structure according to criterion A. is satisfactory when the real thermal resistance of the pavement $R_r$ (m$^2$·K·W$^{-1}$) is equal to or greater than the necessary thermal resistance of the $R_n$ (m$^2$·K·W$^{-1}$) determined based on the requirement to not allow greater freezing of the soil in the subsoil than is allowed. The condition for $n$

pavement construction layers with thickness $h_i$ (m) and thermal conductivity coefficient $\lambda_i$ $(W\cdot m^{-1}\cdot K^{-1})$ is expressed by the formula:

$$\sum_{i=1}^{n} \frac{h_i}{\lambda_i} \geq 0.102 \cdot FI_n^{0.3} + \frac{h_{pf}}{\lambda_{ss}} \tag{1}$$

|  |  |  |  |
|---|---|---|---|
| where: | $FI_n$ | frost index determined for the respective periodicity $n$ | (°C) |
|  | $h_{pf}$ | permissible thickness of pavement freezing | (m) |
|  | $\lambda$ | thermal conductivity coefficient of subgrade soil. | $(W\cdot m^{-1}\cdot K^{-1})$ |

Criterium B. for the assessment of mechanical efficiency of asphalt pavements is described in detail in [17,18] and CCP in [57,58]. According to [39], implemented through TP 098, the structural safety design of cement concrete (CC) cover must satisfy the condition that the maximum flexural tensile stress from a single load and repeated loads on the longitudinal or transverse joint of the CC plate must be greater or equal than to the flexural tensile strength of the cement concrete at the specified utilization factor. When calculating the stresses in the CC plate, the joint effect of stresses from traffic and thermal stress is assessed. According to TP 098 for the pre-assessment of CCP can be used modified Westergaard equations, the rigid plate calculation on elastic multi-layered half-space, Pickett, and Ray influence surfaces, and for the final assessment it is necessary to use FEM calculation. The method of thermal stress calculation is detailed in [57], where the research results presented in the Conclusions of this article are presented.

*2.3. Dependence of Climatic Characteristics of Central Europe on Altitude*

To design pavements, the following climatic characteristics are used in Slovak Republic conditions:

- Average annual air temperature-design of concrete pavements;
- Frost index (FI)-design of asphalt and concrete pavements.

These characteristics are obtained by evaluating air temperature measurements. According to international conventions, this temperature is measured at a height of 2.0 m above terrain at 7 a.m., 2 p.m., and 9 p.m. during the day. For practice, the daily temperature flow is expressed by average daily air temperature $T_s$, which is calculated as:

$$T_s = (T_7 + T_{14} + 2\cdot T_{21})/4. \tag{2}$$

The average annual air temperature $T_a$ (°C) is expressed by the following formula:

$$T_a = \sum_{i=1}^{365} T_{s,i}/365. \tag{3}$$

In Slovakia, the civil engineering design value $T_a$ is determined from long-term measurements of air temperatures (Figure 3, selected meteorological stations in Slovakia), for pavement engineering purposes is used the average annual temperature map according to STN 73 6114 [59] or objectified research results.

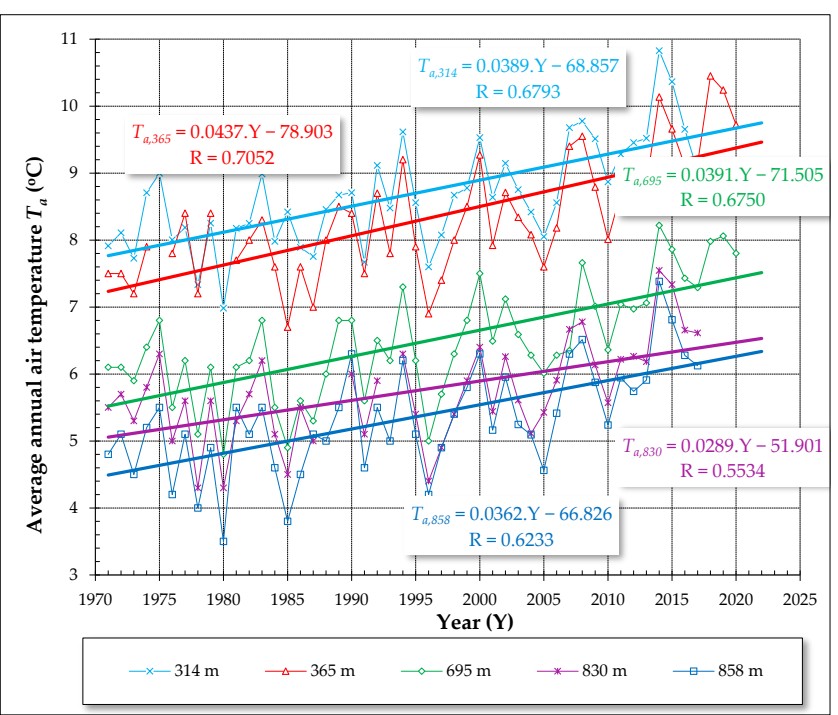

**Figure 3.** Correlation dependencies of the evolution of annual average temperatures for altitudes 314, 365, 695, 830, and 858 m a.s.l. for the period 1971 to 2020.

The following equations for the computing of $T_a$ were objectified for the $SL$ above 300 m and individually evaluated time periods A (Figure 4):

• Average annual temperature 1971–2000;

$$T_{a,1971-2000} = -0.00574 \cdot SL + 10.06 \qquad (4)$$

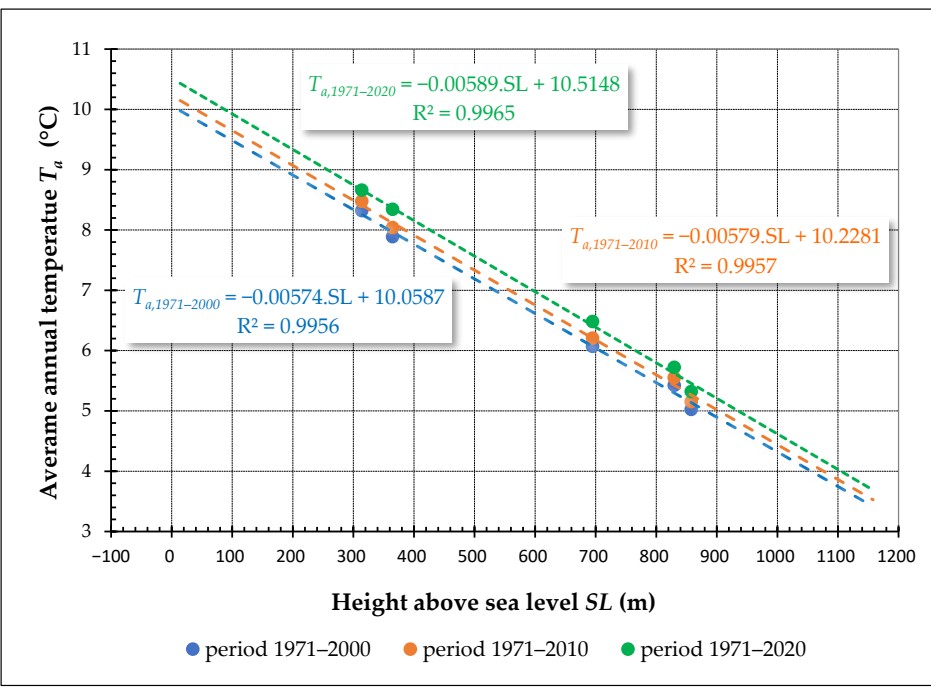

**Figure 4.** Dependence of the average annual air temperature of Central Europe for the period 1971–2000, 1971–2010, and 1971–2020 exceeding 300 m above sea level.

- Average annual temperature 1971–2010;

$$T_{a,1971-2010} = -0.00579 \cdot SL + 10.23 \tag{5}$$

- Average annual temperature 1971–2020;

$$T_{a,1971-2020} = -0.00589 \cdot SL + 10.51. \tag{6}$$

In a scientific monograph [60] and contributions [18,38] was presented the correlation of frost indexes *FI* (as the sum of the negative average daily air temperatures consecutive in winter) dependence on the altitude of specific meteorological stations of Slovakia, starting from the logical premise of frost index dependence on altitude (*SL*).

The equations presented in Figure 5 express the dependence of the computing frost indexes of $FI_n$ (for different periodicity *n* of 0.10, 0.15, and 0.25) which were objectified for the *SL* above 300 m and individually evaluated time periods.

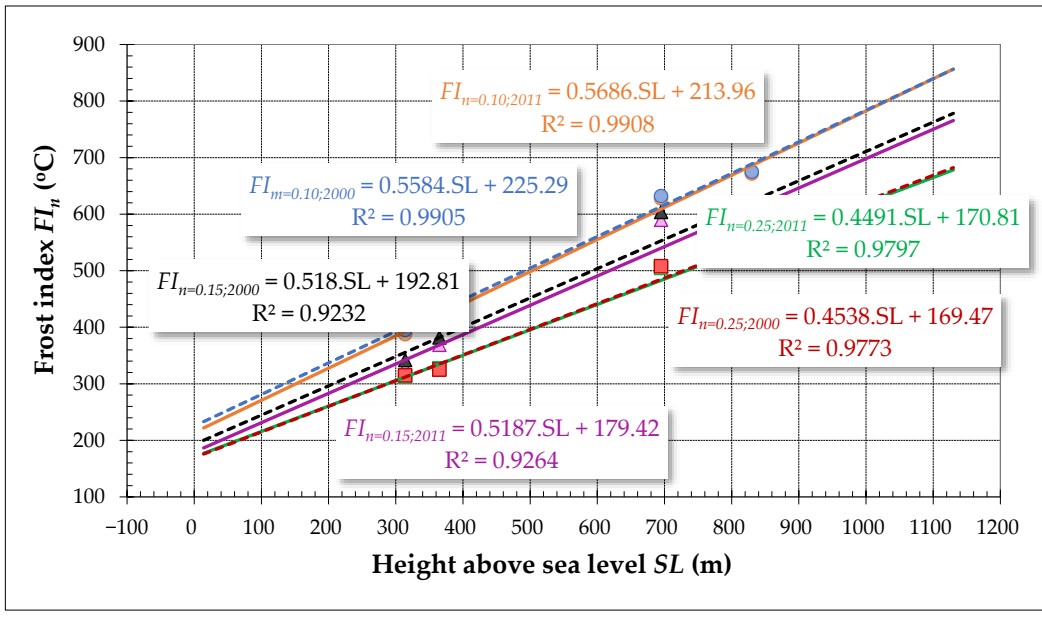

**Figure 5.** Objectified correlation dependences of the Slovak design values of the frost index $FI_n$ for the periodicity 0.10, 0.15, 0.25 on the altitude *SL* for the period 1971 to 2000 and 1971 to 2011.

The authors conducted separate evaluations of the dependence of frost indices on the elevation of the pavement of interest for the period 1971 to 2000 and the period 1971 to 2011. By comparing the objectified $FI_n$ dependencies for all periodicities (*n* of 0.10, 0.15, 0.25), an average level of 98.4% of the RI values determined for the period 1971 to 2000 was obtained for the period 1971 to 2011 evaluated. This represents a 1.6% reduction in FI when considering the average values for the 40 years (1971–2011) and the 30 years from 1971 to 2000.

### 2.4. Flexural Strength of Isomorphic Models of Foam Concrete FC 500 Construction Layer

Two isomorphic models of base systems for traffic structures with a base layer of foam concrete were built at the authors' workplace, namely the model of FC 500 base layer and the model with FC 500 base layer with a reinforcing basalt mesh ORLITECH® Mesh (OM). In previous work by the authors [48–50], the possibility of using composite models of foam concrete with nonwoven PP (polypropylene) geotextile 200 g·m⁻² (Filtek); geogrid with nonwoven PP fibers 60 g·m⁻², and PET (polyethylene terephthalate) mesh (Armatex); and geogrid made of stretched monolithic flat bars and filter geotextile (Combigrid) have been verified in situ and in the lab. The best result was obtained with the nonwoven polypropylene geotextile material, which was used for the applications reported in this paper.

The basalt (composite) reinforcing mesh was chosen as the optimal replacement for steel reinforcement for its advantages such as lack of corrosion compared to steel mesh, lower weight (density of 360 g·m$^{-2}$), higher tensile strength (1284 to 1458 MPa), better cohesion with concrete, lower thermal expansion (thermal coefficient of linear expansion of 6 to 10 × 10$^{-6}$ °C$^{-1}$), fewer cracks in concrete, thermal and electrical insulator, as well as easy handling and installation. The composite mesh used consisted of bars in thickness of 3 mm in diameter and 100 × 100 mm mesh size, in 2 perpendicular directions connected by special material. The application of OM is also very advantageous in marine environments such as harbors, piers, and dikes and in structures that need electromagnetic neutrality such as hospitals. The measurements of foam concrete models were performed in-situ and were complemented by laboratory measurements of mechanical characteristics.

Under laboratory conditions, flexural strength was determined as a material characteristic that is the critical property of FC in terms of reaching the limit state of pavement use and is input into the design and assessment calculations of pavements of engineering structures. The strength was calculated for test specimen FC 500 without geotextile, FC 500 with standard nonwoven PP geotextile GTX 200 g·m$^{-2}$, and FC 500 with GTX 200 with basalt mesh. The test specimen was a prism-shaped beam (100 × 100 × 400 mm) in accordance with STN EN 12390-5:2020 [61] and the set contained a total of 49 specimens. The flexural strength $f_{cf}$ of the test specimen was determined by applying a symmetrical two-point method of loading at a constant loading rate on a servo-controlled testing device [61]. As the specimens were subjected to bending, the maximum load was recorded once the tensile stresses exceeded the flexural strength of the FC and cracks started to occur when the bending moment was reached (Figure 6).

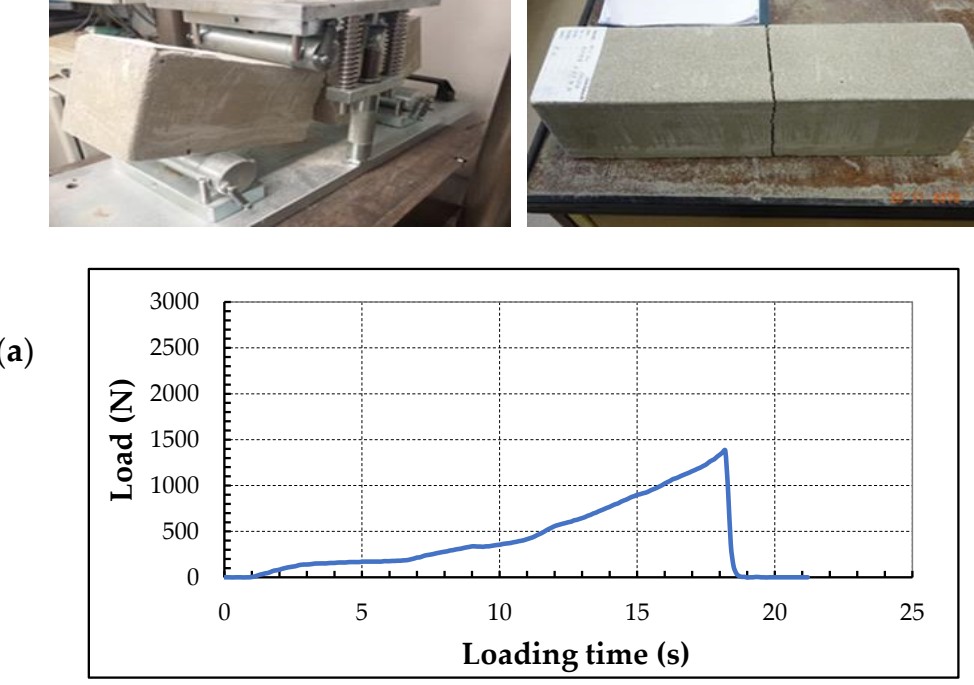

**Figure 6.** *Cont.*

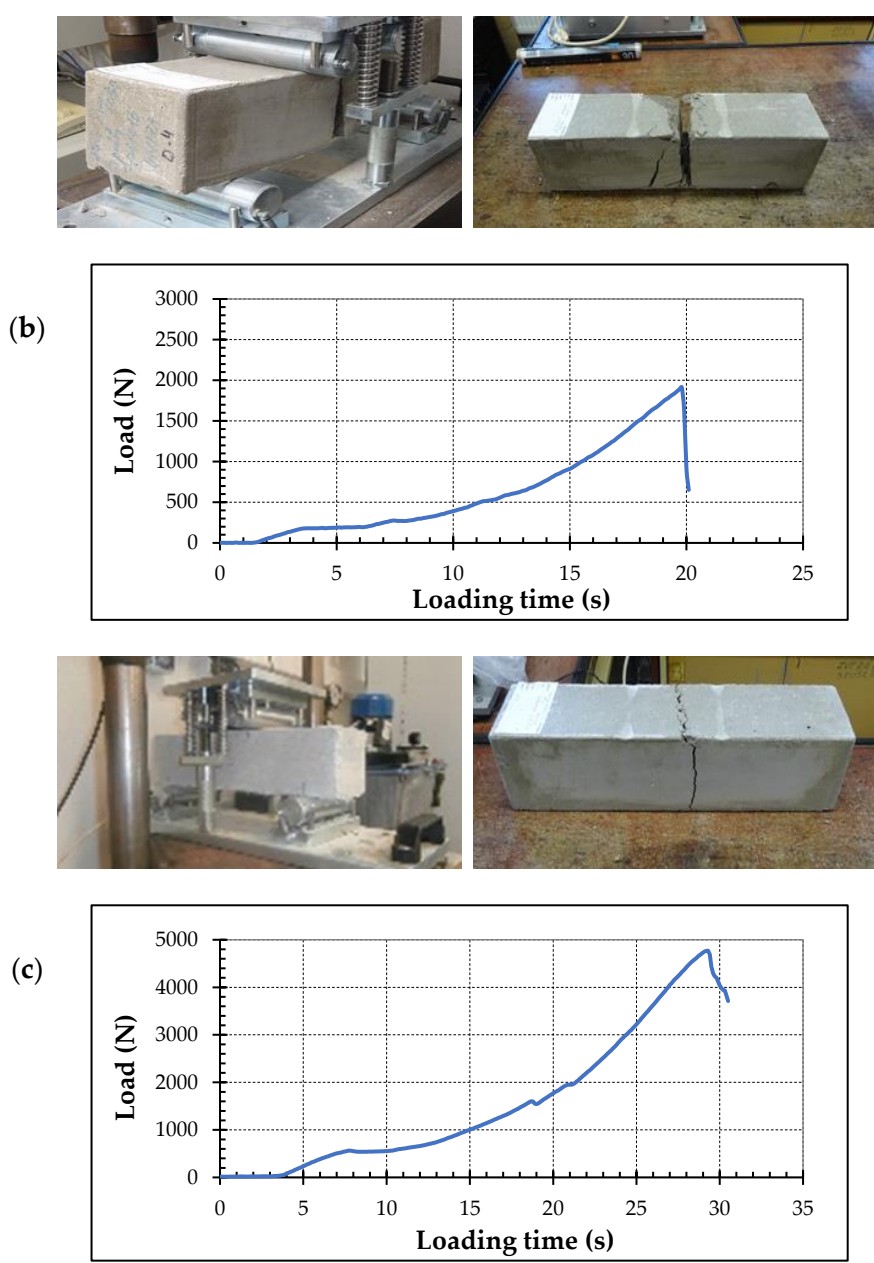

**Figure 6.** Test samples and examples of record of the test; (**a**) foam concrete FC 500, (**b**) foam concrete FC 500 + GTX 200, and (**c**) foam concrete FC 500 + GTX 200 + OM.

The Grubbs test and Dixon's critical values test were used to remove outliers in the set of measurements. The measured flexural strength of the tested specimen after removing outliers is shown in Table 1 and Figure 7. Before the determination of flexural strength, the measurement of density on compaction of the foam concrete was performed (Table 1). The results showed that the use of basalt mesh will significantly increase the flexural strength of foam concrete, as compared to hydraulically cement bound granular mixture CBGM with strength classification $C_{5/6}$ and design value of flexural strength 0.50 MPa which is standardly used in a base layer of pavements. A similar increase in flexural strength can be achieved by using fibers in foam concrete, which has been demonstrated by many measurements [27,34]. To determine the design values of the foam concrete characteristics, the lower limit of the mean value interval at a 95% confidence level was used as input data for the three-dimensional modeling of pavement with a layer of foam concrete.

**Table 1.** Flexural strength and density of tested FC specimens.

| | Density Average Value (kg·m$^{-3}$) | Flexural Strength (N·mm$^{-2}$) | | | |
| --- | --- | --- | --- | --- | --- |
| | | Average Value | Standard Dev. | Student's Distribution $t_{0.05}$ | Design Value |
| FC 500 | 522 | 0.376 | 0.055 | 2.080 | 0.36 |
| FC 500 + GTX 200 | 516 | 0.521 | 0.086 | 2.069 | 0.48 |
| FC 500 + GTX 200 + OM | 525 | 1.370 | 0.165 | 4.303 | 0.96 |

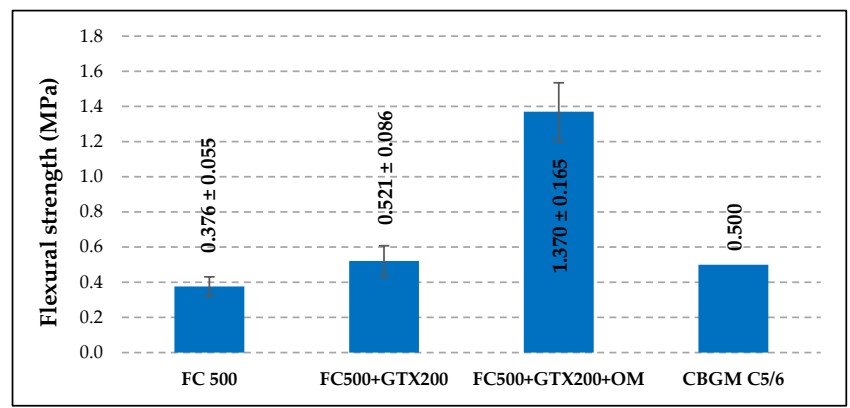

**Figure 7.** Results of flexural strength of tested foam concrete samples tested.

The isomorphic models of the base systems of the traffic structures were built on a uniform subbase formed by anthropogenesis soil of intermediate-plastic clay of rigid consistency (Proctor compaction test, optimum water content 15.8% and bulk density 1632 kg·m$^{-3}$), which is often found in the territory of the Slovak Republic. The bearing capacity determined by a static load test was determined to be 15.0 MPa.

To determine the properties of different load-bearing bases, both composite models were built on two different sub-base layers. The first sub-base layer was constructed of coarse aggregate 8/16 mm in 150 mm thickness separated from the subgrade by a separating geotextile. The bearing capacity of the sub-base layer was 19.6 MPa (Figure 8). The second sub-base structure was constructed of a 350 mm thick coarse aggregate layer on the subgrade using a separation geotextile with a bearing capacity of 33.3 MPa after compaction. Two isomorphic models of the base system from foam concrete FC 500 with GTX 200 and composite FC 500 with GTX 200 reinforced by basalt mesh in 220 mm thickness were made on both sub-base constructions (Figure 9).

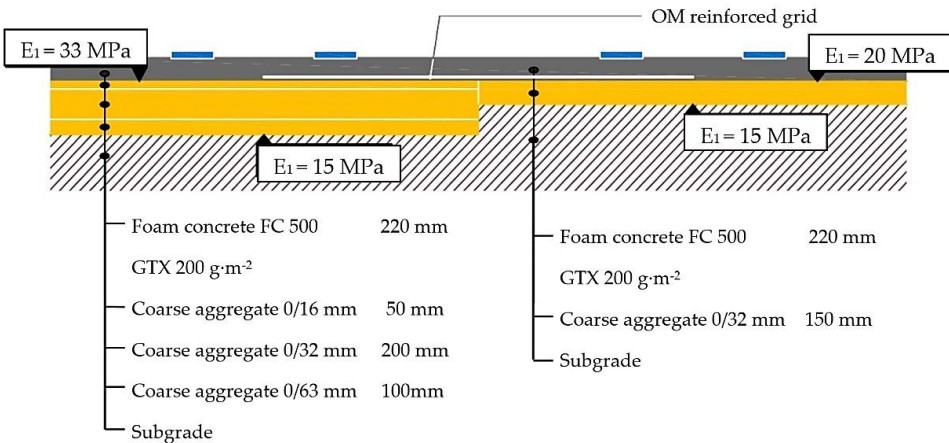

**Figure 8.** Isomorphic models of base construction.

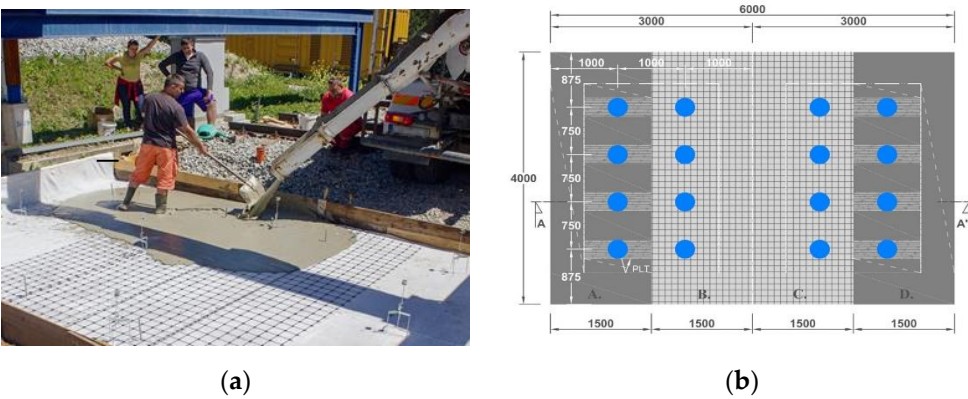

**Figure 9.** (**a**) Construction of experimental isomorphic models, (**b**) scheme of measuring points.

The maximum contact stress of loading by the static load test was chosen to be 0.6 MN·m$^{-2}$ corresponding to the design vehicle axle parameter with a mass of 10 tones. The achieved load-bearing capacity values expressed by the modulus of elasticity from the second loading cycle $E_e$ are shown in Table 2, from which the modulus of elasticity of the base layer from foam concrete (FC 500 and FC 500 + OM) was subsequently determined by back-calculation from the measured values of the equivalent modulus of elasticity on the surface of subgrade and the sub-base layer and known thicknesses (Table 2). Similar to the results of the laboratory measurements (Table 1), the in-situ measurements showed that reinforcing the foam concrete with basalt mesh will increase the modulus of elasticity of the foam concrete layer. The results obtained are comparable to the values found by [62,63] where for foam concrete densities of 500 to 1600 kg·m$^{-3}$ the values of the modulus of elasticity are in the range of 1 to 12 kN·mm$^{-2}$.

**Table 2.** Results of the modulus of elasticity.

| Isomorphic Model | Modulus of Elasticity (MPa) | | |
|---|---|---|---|
| | Sub-Base | Equivalent Modulus $E_e$ | Back-Calculation $E$ |
| FC 500 + GTX 200 | 19.6 | 82.50 | 1412.5 |
| | 33.3 | 130.75 | 2062.5 |
| FC 500 + GTX 200 + OM | 19.6 | 107.75 | 3675.0 |
| | 33.3 | 155.00 | 3575.0 |

## 3. Three-Dimensional Model of a Cycle Pavement with a Layer of FC Reinforced with a Basalt Mesh

The design of the pavement system is influenced by axle loads, axle configuration, tire contact patches, number of load cycles, vehicle speed, and the effect of temperature changes on the pavement or area declination. Understanding the constitutive relationship between stress and strain in FEM is more than necessary. This is due to the use of stress-strain analysis to predict pavement failures. Thus, the relative conditions of different layers in the pavement structure can also be analyzed. Some basic issues about pavement performance can be best answered by three-dimensional finite element analysis tools. However, the processing and time required to model pavements accurately make these analyses difficult to use [64]. The traffic means (vehicles) are considered as static load. Dynamic load is still the subject of research.

In accordance with the Road Act No. 135/1961 [39], when assessing the final design of a cement concrete pavement (CCP), stresses must be evaluated using FEM based on the limit axle load 2P = 115 kN, in accordance with the requirements of TP 098 [58]. FEM is a universal and very efficient numerical method for solving continuum mechanics problems. In general, it involves dividing the solved continuum surface into a system of small finite elements, connected at the nodes of the mesh. The generated discrete system must satisfy

the continuity and the balance conditions. The created model can be made of 1D, 2D, and 3D elements, each element of which is described by a system of algebraic equations.

In the case of the computational model of the cycle pavement (Pavement 3) with a layer of foam concrete (FC 500) reinforced with a basalt mesh-ORLITECH composite system (OM), it is a surface stress solution. It is a complicated model created from 3D elements in VisualFEA (8 node quadrilateral) providing all the necessary information about the stiffness element, which it includes in the elements of the stiffness matrix. The finite elements used and the process of creating the 3D model is shown in Figure 10.

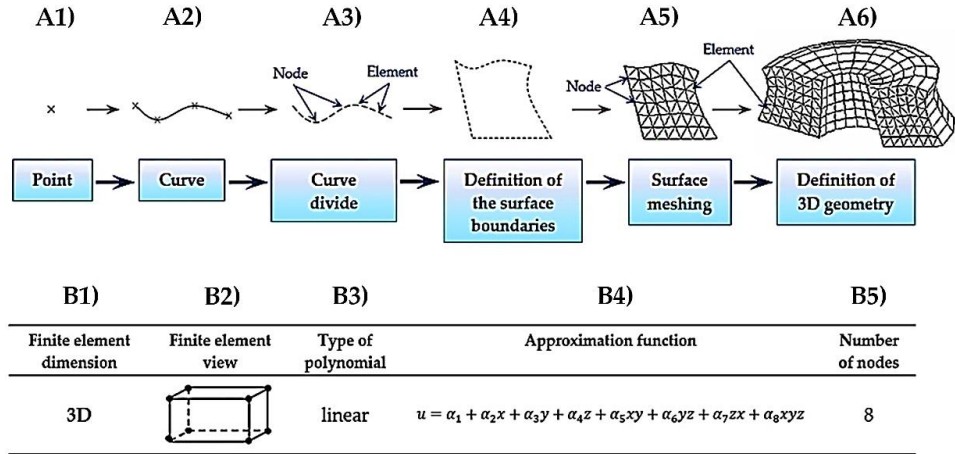

**Figure 10.** Finite elements used in modeling and the process of creating a 3D model; (**A1**)—definition of basic points of model shape; (**A2**)—connecting basic points using lines and curves (thread model simplified); (**A3**)—dividing lines and curves (setting boundaries for elements); (**A4**)—definition which lines creates boundaries for element meshing procedure (surface mesh); (**A5**)—meshing of defined surfaces with 2D element types (not assigned width); (**A6**)—from defined surface boundaries creating of volume element mesh (or other advanced methods can be used); (**B1**)—real dimension of element used for pavement layer; (**B2**)—hexahedron type of volume element used for model; (**B3**)—type of polynomic interpolation for used element; (**B4**)—mathematical expression of approximation function for FEM solution; (**B5**)—number of element nodes for solving of displacement (define the difficulty of computation).

The layers of the numerical model of the cycle pavement and the mechanical characteristics used are shown in Table 3 (Pavement 3).

**Table 3.** Cycle pavement construction and characteristics of layers.

| Cycle Pavement Construction | Thickness (mm) | Modulus of Elasticity (MPa) | | | Poisson's Ratio (MPa) | | | Flexural Tensile Strength (MPa) | | |
|---|---|---|---|---|---|---|---|---|---|---|
| | | 0 °C | 11 °C | 27 °C | 0 °C | 11 °C | 27 °C | 0 °C | 11 °C | 27 °C |
| Single-layer cement concrete pavement CC III; PE (polyethylene) geotextile | 180 | | $35.10^3$ | | | 0.2 | | | 4.0 | |
| Foam concrete with reinforcing mesh FC 500 + OM, base layer Gravel crushed stone | 100 | | 1800 | | | 0.23 | | | 1.0 | |
| Unbound gravel materials (UGM), sub-base layer | 150 | | 350 | | | 0.3 | | | - | |

The influence of the layers (except the stiffest layer), including the subgrade, is included by a linearized calculation in the Spring constant per unit area $\Delta z$ of the 3D element. In this case, the value of $\Delta z = 34.34$ MN·m/m$^2$ is obtained by substituting the modulus of elasticity

of the subgrade $E = 30$ MPa, the effective width of the contact area of the rigid elements with the elastic subgrade $B = 0.96$ m, and the Poisson number $\nu = 0.3$ into relation (7).

$$\Delta_z = E / \left[ B \cdot \left( 1 - \nu^2 \right) \right] \tag{7}$$

The contact between the main layers (CC III and FC 500 + OM) had to be solved by using a special tool of VisualFEA-Volume Interface, which can reduce the perfectly rigid interaction of the layers on the contact surfaces. The influence of the PE separation geotextile is thus considered. The reinforcing 1D elements of the basalt mesh are modeled using the Embedded Bar function and the results from the simplified model are shown in Figure 11. Future work will continue to refine the 3D model and experimental measurements will be processed to calibrate and verify the developed model.

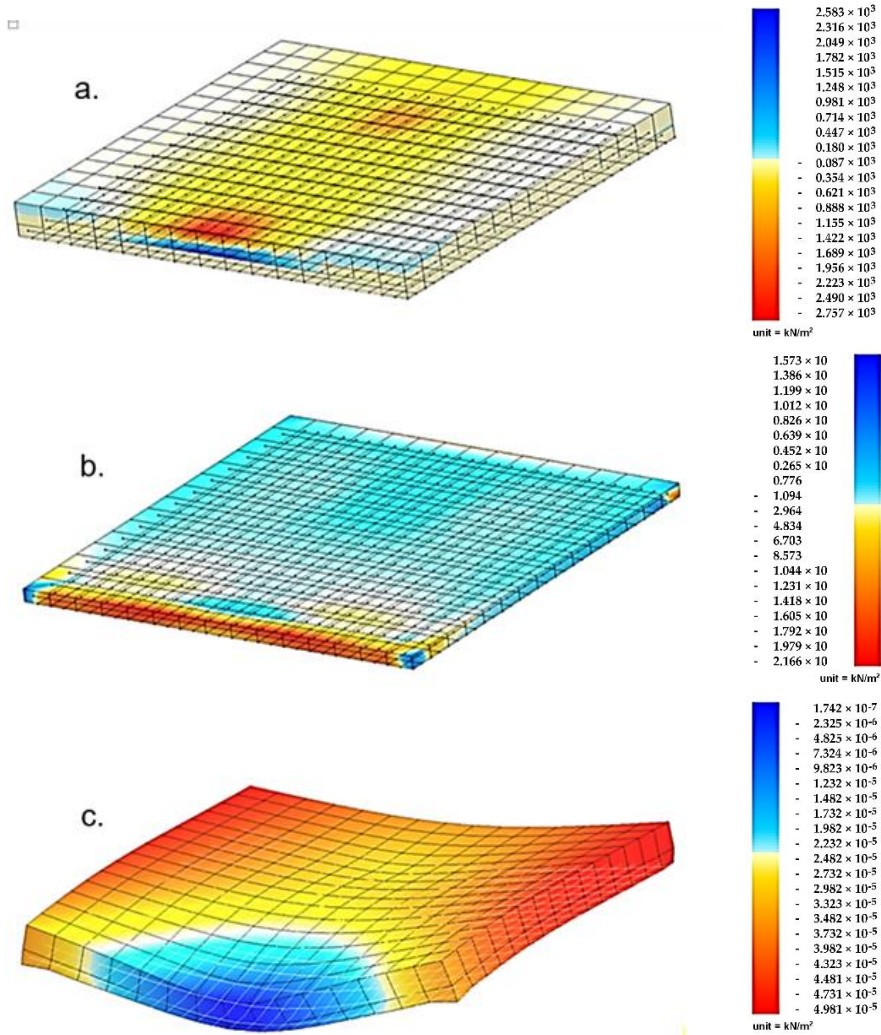

**Figure 11.** Skin stress and strain on the FEM model, (**a**) maximum normal stresses on the model $\sigma_x$, (**b**) maximum normal stresses $\sigma_x$ on the 3D foam concrete solid, (**c**) maximum displacement from the 2P load.

A numerical model of the pavement was created from the solid elements. In this model, it is possible to track not only the stresses on the surfaces (skin stresses) but also the changes along with the height of the solid.

The plate models do not allow this option (Figure 11a). Another option is to turn off the particular level at which the pavement layers are modeled. It is then possible to watch the effect of the load, for example on a layer of foam concrete (Figure 11b). The

maximum deformation in the vertical direction due to load as required by TP 098 is shown in (Figure 11c).

Because dimensional stresses are solved, it is also possible to obtain from the model the contact stresses arising between the modeled parts. According to TP 098 for the pre-design of dimensions and design assessment in the project documentation for the construction plan or planning permission, the modified equations can also be used. These are the Westergaard equations (WEST), the rigid plate calculation on elastic multilayered half-space (LAYMED) or the Pickett and Ray influence surfaces (PICRA), and Table 4 shows the stresses obtained by these methods.

**Table 4.** Stresses results obtained by different methods.

| Numerical Results-Stresses (Obtained by Different Methods) | Model of Cycle Pavement 3 | Single Load |
|---|---|---|
| Westergaard | (WEST) | 3.113 MPa |
| Pickett and Ray | (PICRA) | 3.298 MPa |
| Elastic multilayered half-space | (LAYMED) | 2.965 MPa |
| Finite element method | (FEM) | 2.583 MPa |
| Thermal Stresses | (THERM) | 1.083 MPa |
| Criterial Limit value according to standard TP 098 | CC III | 4.0 MPa |

The authors addressed the method of determining thermal stresses in road constructions in the following scientific paper. Decky, M., Papanova, Z., Juhas, M., and Kudelcikova, M. (2022). Evaluation of the Effect of Average Annual Temperatures in Slovakia between 1971 and 2020 on Stresses in Rigid Pavements. *Land*, *11*(6), 764 [57]. One of the main conclusions [57] is that using composite foam concrete is most appropriate in terms of mitigating the effects of temperature on cement concrete pavements, especially on jointed plain concrete pavements and jointed reinforced concrete pavements.

## 4. Discussion

The current civil engineering industry is a dominant contributor to carbon dioxide emission [65] and therefore is necessary to reduce it through the circular economy pavement circular economy (PCE). Generally, CE is perceived as an industrial economy that is restorative or regenerative by intention and which aims to keep products, components, and materials at their highest utility and value, all the time [66–68]. It is necessary to try to choose more environmentally friendly variants of construction projects, and use a more environmentally friendly material, which has a higher degree of recyclability and reusability [69,70]. The European experts in the field of PCE ascribe the highest importance to the high durability of asphalt and cement concrete pavements (perpetual), special innovative material design to meet climatic requirements, and wider usage of recyclable materials [70–72]. A recent increase in the use of ecofriendly, natural fibers as reinforcement for the fabrication of lightweight, low-cost polymer composites can be seen globally. One such material of interest currently being extensively used is basalt fiber, which is cost-effective and offers exceptional properties over glass fibers [73]. Despite the many benefits of using basalt fiber in concrete, only a restricted number of research has been found in the literature concerning basalt fiber or mesh reinforced construction layer of pavements [74–77]. The authors' long-term research of climatic changes of CE [17–19,57,60] and research in the field of applicability of foam concrete with a bulk density of 500 kg·m$^{-3}$ in transport constructions [48–50] created credibility conditions for the design cycle pavements using composite foam concrete (CFC) with basalt mesh at high altitudes of CE. In the area of increasing the mechanical efficiency of the road, the increase in the tensile strength of CFC from 0.5 MPa to 1.3 MPa was objectified due to the use of a basalt reinforcement net. To provide credible design and assessment of such pavements through validated FEM models, they present

the results of research on climatic characteristics for the period 1971–2020, according to the World Meteorological Organization, the weather trends for the last 30 years are assessed.

As part of their research, the authors objectified the correlation dependences of average annual temperatures and frost index from the altitudes of Central Europe exceeding 300 m above sea level (Figures 4 and 5). The following results from Figure 4 determine the evaluated time intervals and the considered altitudes of 300 and 1100 m above sea level are:

- Average annual temperature 1971–2000 8.34 °C (300 m) and 3.75 °C (1100 m);
- Average annual temperature 1971–2010 8.49 °C (300 m) and 3.86 °C (1100 m);
- Average annual temperature 1971–2020 8.75 °C (300 m) and 4.03 °C (1100 m).

Due to increasing air temperatures, the frost index decreases and for the above range of altitudes, the authors found:

- Average value of the frost index in the period 1971–2000 for:

  $n = 0.10$ 392.8 (300 m) and 839.5 (1100 m);
  $n = 0.15$ 348.2 (300 m) and 762.6 (1100 m);
  $n = 0.25$ 305.6 (300 m) and 668.7 (1100 m);

- Average value of the frost index in the period 1971–2011 for:

  $n = 0.10$ 384.5 (300 m) and 839.4 (1100 m);
  $n = 0.15$ 335.0 (300 m) and 750.0 (1100 m);
  $n = 0.25$ 305.5 (300 m) and 664.8 (1100 m).

Using the presented research results, which resulted in the granting of Slovak patents for the described CFC structural layer, the following structural composition of the cycle pavements was optimized. For the possibility of mutual comparison and interchangeability of the pavement, covers were designed rigid (1–3), semi-rigid (4 and 5) pavements, whereas pavement 3 and 5 has innovative base courses CFC from foam concrete FC 500 reinforced by basalt mesh.

The results in Table 5 show the possibilities of reducing the overall pavement thickness by 40 mm for the rigid pavement with a cement concrete surface and by 100 mm for the semi-rigid pavement with an asphalt surface. In continuing the research, the authors want to focus on further saving non-renewable natural resources and assessing of life cycle cost of CFC and their environmental impact.

**Table 5.** The overview of thickness (mm) cycle pavement construction layers of Pavements 1 to 5.

| Pavement Construction Layer | Pavement 1 | Pavement 2 | Pavement 3 | Pavement 4 | Pavement 5 |
|---|---|---|---|---|---|
| Cement concrete for surface CC III; EN 13877 | 190 | 190 | 180 | - | - |
| Asphalt concrete for base course AC 11; EN 13108-1 | 40 | - | - | - | - |
| Asphalt concrete for wearing course AC 8; EN 13108-1 | - | - | - | 30 | 30 |
| Asphalt concrete for binder course AC 16; EN 13108-1 | - | - | - | 50 | 50 |
| Asphalt concrete for road base AC 16; EN 13108-1 | - | - | - | 50 | 50 |
| Cement bound granular mixture CBGM $C_{5/6}$; EN 14227-1 | 150 | - | - | 150 | - |
| Foam concrete + reinforcing basalt mesh FC 500 + OM | - | - | 100 | - | 100 |
| Unbound mixtures (UM) for sub-base layer UM 0/31.5; EN 13285 | - | 180 | - | - | - |
| Unbound mixtures (UM), for sub-base layer UM 0/31.5; EN 13285 | 180 | 180 | 150 | 200 | 150 |
| Elasticity modulus of subgrade *E* (MPa) | | | min. 30 | | |
| Pavement service life (years) | | | min. 30 | | |
| Total pavement thickness (mm) | 560 | 550 | 430 | 480 | 380 |
| The required thermal resistance of the pavement (m²·K·W⁻¹) | | | 0.260 for altitude 300 m according to TP 098 [58] 0.412 for altitude 1100 m according to TP 098 [58] | | |
| The real thermal resistance of the pavement (m²·K·W⁻¹) | 0.263 | 0.285 | 0.650 | 0.286 | 0.676 |

## 5. Conclusions

With the increase in global warming, the construction sector [17,18,57] is trying to find an alternative to ordinary concrete and presently the emerging trend is the use of foamed concrete, which is a lightweight concrete having a greater strength-to-weight ratio with density varying from 300 to 1800 kg·m$^{-3}$ [33,63]. To use in the base layers of road structures, the need to increase its tensile strength in bending was identified [27,62]. The authors have been intensively addressing these research issues for the last 10 years [48,49] and in this article, they present the application of composite concrete with a bulk density of 500 kg·m$^{-3}$, reinforced with a basalt mesh [50].

The authors present an optimized pavement design of bicycle paths with base layers made of composite foam concrete. First, it was necessary to develop a reliable method for determining the climatic characteristics of Central Europe (CE), specifically the average annual temperature $T_a$ and the frost index (FI). For CE altitudes from 300 to 1100 m and the evaluated period 1971–2020, the temperature range is 8.8 to 4.0 °C. To illustrate the significant effect of $T_a$ on thermal stresses (TS) of cement concrete plates, we give corresponding numerical values for its thickness of 20 cm and dimensions of CC plates 4 × 4 m for temperatures $T_a$ = 4.5 and 11.5 °C and modulus of reaction $k$ [57]:

- k = 100 MN·m$^{-3}$　　　　$TS_{4.5}$ = 2.26 MPa　　　　$TS_{11.5}$ = 1.64 MPa;

- k = 200 MN·m$^{-3}$　　　　$TS_{4.5}$ = 2.63 MPa　　　　$TS_{11.5}$ = 1.91 MPa;

- k = 300 MN·m$^{-3}$　　　　$TS_{4.5}$ = 2.72 MPa　　　　$TS_{11.5}$ = 1.92 MPa.

For the evaluated period 1971–2011 and periodicity n = 0.10, 0.15, 0.25, this range is from $FIn$ = 385, 335, 306 °C to 839, 750, 665 °C. The evaluation from the point of view of the design of the road confirmed for Central Europe a significant increase in temperatures over the last 10 years and a very slight decrease in frost indices.

When assessing the mechanical efficiency of pavement construction, the decisive characteristic is the flexural strength of the subbase layers. Through extensive experiments on homomorphic [20,78] and isomorphic models [49,79] of pavement layers, the authors objectified the following research results. For foam concrete with a volume weight of 500 kg·m$^{-3}$ (FC 500) with the use of geotextile (GTX) and reinforcing basalt Orlitech mesh (OM), the following average flexural strengths were objectified: FC 500 = 0.38 MPa, FC 500 + GTX = 0.52 MPa, FC 500 + GTX + OM = 1.37 MPa.

The determined strengths are comparable to the standard subbase material as CBGM (Cement Bound Granular Mixture) with strength classification $C_{5/6}$, $C_{8/10}$, and $C_{12/15}$, which have design values of flexural strength of 0.50, 0.80, and 1.00 MPa [59].

The presented research results made it possible to reduce the total thickness of cement concrete cycling pavements (CCCP) from 56 to 43 cm and flexible (asphalt) concrete from 48 to 38 cm for the high altitudes of Central Europe (Table 5). The validated FEM model is required for final assessments of rigid pavements by the requirements of Act No. 135/1961, Act on Roads [39] implemented to pavement design through TP 098 [58].

**Author Contributions:** Conceptualization, M.D.; Formal analysis, Z.P.; Investigation, E.R.; Methodology, E.R.; Software, Z.P.; Supervision, M.D.; Visualization, K.H.; Writing—original draft, M.D., K.H., Z.P. and E.R. All authors have read and agreed to the published version of the manuscript.

**Funding:** This research was funded by Ministry of Education, Science, Research and Sport of the Slovak Republic, grant number KEGA:027ŽU-4/2022.

**Institutional Review Board Statement:** Not applicable.

**Informed Consent Statement:** Not applicable.

**Data Availability Statement:** Not applicable.

**Conflicts of Interest:** The authors declare no conflict of interest.

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
