# Peer review of "Sustainable Adaptive Cycle Pavements Using Composite Foam Concrete at High Altitudes in Central Europe"

_sustainability, doi:10.3390/su14159034_

Round 1

Reviewer 1 Report

This manuscript was well written titled as “Sustainable adaptive cycle pavements using composite foam concrete at high altitudes in Central Europe”. This manuscript was well written and presented. However, there are several critical doubts about the contents, please carefully consider. Thanks.

1. In lines 187 to 199, the index ‘Im,n, what does ‘m’ and ‘n’ represent? How does this index work in this study? And what is the relationship with the front and rear frontal contents?

2. In line 199, ‘the values obtained for the period 1971 to 2011 (Figure 5) reached a rank of 98.4 %’, the value of 98.4% appeared suddenly, it was hard to grasp its logical and intention.

3. The results in Figures 6, 7, and 8 was not consistent with that in Table 1, especially the flexural strength of FC500 and FC500 + GTX200, please verify.

4. In line 228, ‘in accordance with EN 12360-5 [55] and the set contained a total of 49 specimens’, ‘EN 12360-5’ is not correct, please modified.

5. There are lots of abbreviations in this manuscript, but some are not clear for the readers. Please explain them one by one in an appropriate site, such as CB III, PE, UM SD, B, etc.

6. In line 314, ‘Pavement 3’ appeared suddenly.

7. In line 347, ‘thermal stresses appeared suddenly, there are not relevant explanations for the thermal analyses, the thermal parameters and analysis step and boundary, etc. This need critical verify.

8. The conclusion was a little long, the main ideas were not clear, please refine them.

9. In line 436, the conclusion of ‘At present, the standard subbase materials are CBGM (Cement Bound Granular Mixture) C5/6, C8/10 and C12/15, which have design values of flexural strength of 0.50, 0.80 and 1.00 MPa.’, the reason of this conclusion was short in the front contents.

10. In line 439, the conclusion of ‘Validated FEM models of cycle pavements showed a reduction in total pavements thickness from 56 to 38 cm for rigid pavements and 48 to 38 cm for flexible pavements (Table 5)’, there not relevant explanation to support this conclusion from the FEM model in this manuscript.

11. This manuscript made a lot work on the definition of Central Europe and different altitudes site. The relationship between sea level and 17 frost index was obtained. However, the relationship between the sea level and stress or strength of foam concrete was not clear from this study. These are individual parts between section 2.1, 2.2 and section 2.3 from my view. 

Reviewer 2 Report

Manuscript ID: sustainability-1788788

Manuscript title: Sustainable adaptive cycle pavements using composite foam concrete at high altitudes in Central Europe

General comments:

This work is generally well written, there is a very clear logistical line embedded in the manuscript, which is great. It can be accepted after all my comments are well addressed.

Specific comments:

1.      Abstract. There should be the research results introduced in abstract. Authors introduced the method too much.

2.      Fig.2(a) is not clear enough for publishing.

3.      Fig.3. The labels of each height should preferably be arranged in sequence.

4.      Fig.4 what is the meaning of these three colors presenting?

5.      Fig.6 – Fig.8 could be combined by organizing these images since they have the same pattern.

6.      Fig.12 is not clear enough.

7.      I think there is some details in experiments and simulation are missed, please consider add them in.

Reviewer 3 Report

1. In the Abstract add meaning of FEM?

2. Figure 2. text (b) to a new row.

3. In Figure 4. move 0C to the top row as in figure 3.

4. Text rows 182-186 above figure 4.

5. In figure 5. Frost index (Im) in °C? 

6. Text rows 190-199 move above figure 5.

7. Row 214: 6°C  is correct.

8. Replace the order of Figures 10. and 11.

9. In rows 380-382 after numbers 8.34, 8,49, and 8.75 add °C.

10. It's ok the frost index in table 5. for the total pavement thickness of 380 mm?  

Round 2

Reviewer 1 Report

The authors made great improvements for this manuscript. However, there are still some suggestions for this manuscript, such as:

1. Generally, little references can be found in the conclusion section. If exist, the conclusion was not refine enough. Please consider.

2. Some parameters were lost in Table 5.

3. In lines 393 to399, some explanations were complemented for the thermal stress analyses. However, there statements come from other results. Your finding was not obvious. Please check.

4. Figure 8 needs check.

5. In line 170, what’s the meaning of ‘nl’?
